# A cerebellar substrate for cognition evolved multiple times independently in mammals

Jeroen B Smaers[1,2]*, Alan H Turner[3], Aida Gómez-Robles[4,5], Chet C Sherwood[5]

[1]Department of Anthropology, Stony Brook University, New York, United States; [2]Center for the Advanced Study of Human Paleobiology, Stony Brook University, New York, United States; [3]Department of Anatomical Sciences, Stony Brook University, New York, United States; [4]Department of Genetics, Evolution and Environment, University College London, London, United Kingdom; [5]Department of Anthropology, The George Washington University, Washington, United States

**Abstract** Given that complex behavior evolved multiple times independently in different lineages, a crucial question is whether these independent evolutionary events coincided with modifications to common neural systems. To test this question in mammals, we investigate the lateral cerebellum, a neurobiological system that is novel to mammals, and is associated with higher cognitive functions. We map the evolutionary diversification of the mammalian cerebellum and find that relative volumetric changes of the lateral cerebellar hemispheres (independent of cerebellar size) are correlated with measures of domain-general cognition in primates, and are characterized by a combination of parallel and convergent shifts towards similar levels of expansion in distantly related mammalian lineages. Results suggest that multiple independent evolutionary occurrences of increased behavioral complexity in mammals may at least partly be explained by selection on a common neural system, the cerebellum, which may have been subject to multiple independent neurodevelopmental remodeling events during mammalian evolution.
DOI: https://doi.org/10.7554/eLife.35696.001

*For correspondence:
jeroen.smaers@stonybrook.edu

**Competing interests:** The authors declare that no competing interests exist.

## Introduction

The brain is the anatomical substrate of behavior. In turn, the behaviors of a species are closely linked to the ecological context and evolutionary history of that species. Large-scale evolutionary modifications in the brain therefore provide essential information about the factors that shape species' diversification patterns. Changes in neurobiological features that directly relate to higher-order cognitive capacities are particularly relevant as they underpin adaptive behaviors such as tool manipulation (*Krützen et al., 2005*; *Boesch and Boesch, 1990*), flexible problem solving (*Benson-Amram et al., 2016*), planning for the future (*Raby et al., 2007*; *Osvath and Osvath, 2008*), and sophisticated communication systems (*Janik, 2013*).

Even though it is commonly agreed that instances of intelligent behavior have evolved independently in different lineages of mammals (*Roth and Dicke, 2005*; *Roth, 2015*), it is unclear whether such convergent behavioral abilities arose from modifications of common neural systems or whether lineage-specific contingency has shaped particular brain circuits according to unique socioecological conditions. This uncertainty has led to different perspectives on what defines 'intelligence'. Comparative psychologists describe it as a domain-general problem solving ability that comprises associative-learning. Such 'general intelligence' has been proposed to equip species with the ability to make mental models of the environment, develop actions based on abstract notions of associations between percepts, and to generate goals from current contexts (*Spearman, 1904*; *Duncan et al.,*

**eLife digest** The brains of mammals consist of the same basic structures, but each of these structures varies from one species to the next. A given structure may be larger in one species than another, for example. It may contain different numbers or sizes of cells. It may even have different connections to other brain regions. By comparing individual brain structures between species, we can map how the mammalian brain has evolved.

Smaers et al. have now done this for the cerebellum, a structure at the back of the brain. The mammalian cerebellum consists of three main areas: the vermis, paravermis, and the lateral hemispheres. Smaers et al. show that in apes, dolphins and seals, the lateral hemispheres are unusually large relative to the cerebellum as a whole. This could indicate that these three groups of animals share a common ancestor with enlarged lateral hemispheres. Yet, genetic studies suggest that this is not the case.

Another possibility is that apes, dolphins and seals independently evolved enlarged lateral hemispheres. This may have given rise to a trait that proved beneficial for each of them. But what might this be? Studies in people suggest that the lateral hemispheres help to support some forms of learning. Apes, dolphins and seals are among only a few species of mammal with the ability to learn new calls and vocalizations. The expansion of the lateral cerebellum may therefore have contributed to the evolution of vocal learning, and this may have occurred independently on at least three separate occasions.

Future work should extend this analysis to other cognitive skills, as well as to other species. Bats, for example, would be of particular interest because of their ability to echolocate. Finally, the lateral hemispheres consist of several subregions that play different roles in learning and information processing. Further experiments should explore whether different subregions have increased in size in different species.

DOI: https://doi.org/10.7554/eLife.35696.002

---

*2000*; *Genovesio et al., 2014*). A different view holds that intelligence comprises the aggregate of cognitive modules of special abilities that evolved within a species in response to specific environments (*Barkow et al., 1995*; *Cosmides and Tooby, 1992*). Under this view, intelligence evolved to perform specific computational strategies that are tailored to solve the task demands of ancestrally recurrent adaptive problems (*Cosmides et al., 2010*).

Here, we examine the mammalian cerebellum to address whether convergent evolution of mammalian cognitive capacities are scaffolded by modifications of this common neural system. The cerebellum may be especially informative in uncovering the coevolution of brain structure and cognition for several reasons. First, unlike the more commonly investigated cerebral cortex, the cerebellum's structural, connectional, functional, and developmental anatomy is relatively uniform (*Larsell, 1970*) and is therefore ideal for the comparison of homologous neural circuits across species (*Smaers, 2014a*). Second, the lateral extension of the cerebellum to form distinct hemispheres arose early in mammalian evolution (*Figure 1*), providing the opportunity to investigate the diversification pattern of a newly evolving neural system. Third, through its connectional integration with heteromodal association areas in the cerebrum, the lateral cerebellum has been hypothesized to be involved in the generation of domain-general higher-order models of mental activity. A central working hypothesis in this context is that the cerebellum imposes a type of cognitive control over information processing that consists of automating sequences of thoughts and actions (*Schmahmann, 1997*). This cerebellar-type cognitive control may underpin many aspects associated with 'intelligent' behavior, such as working memory, executive function, and the development of behavioral learning models (*Strick et al., 2009*). Finally, the strong connectional and functional integration between the lateral cerebellar hemisphere and the cerebrum is also evident in developmental modularity. Whereas the medial cerebellum develops early, the lateral cerebellum develops later in tandem with cerebral association areas (*Tiemeier et al., 2010*; *Altman and Bayer, 1997*).

We investigate the extent to which mammalian lateral cerebellar hemispheres evolved in coordination with the rest of the cerebellum, whether they are correlated with measures of domain-general cognitive performance, and what patterns underlie their evolutionary diversification. We primarily

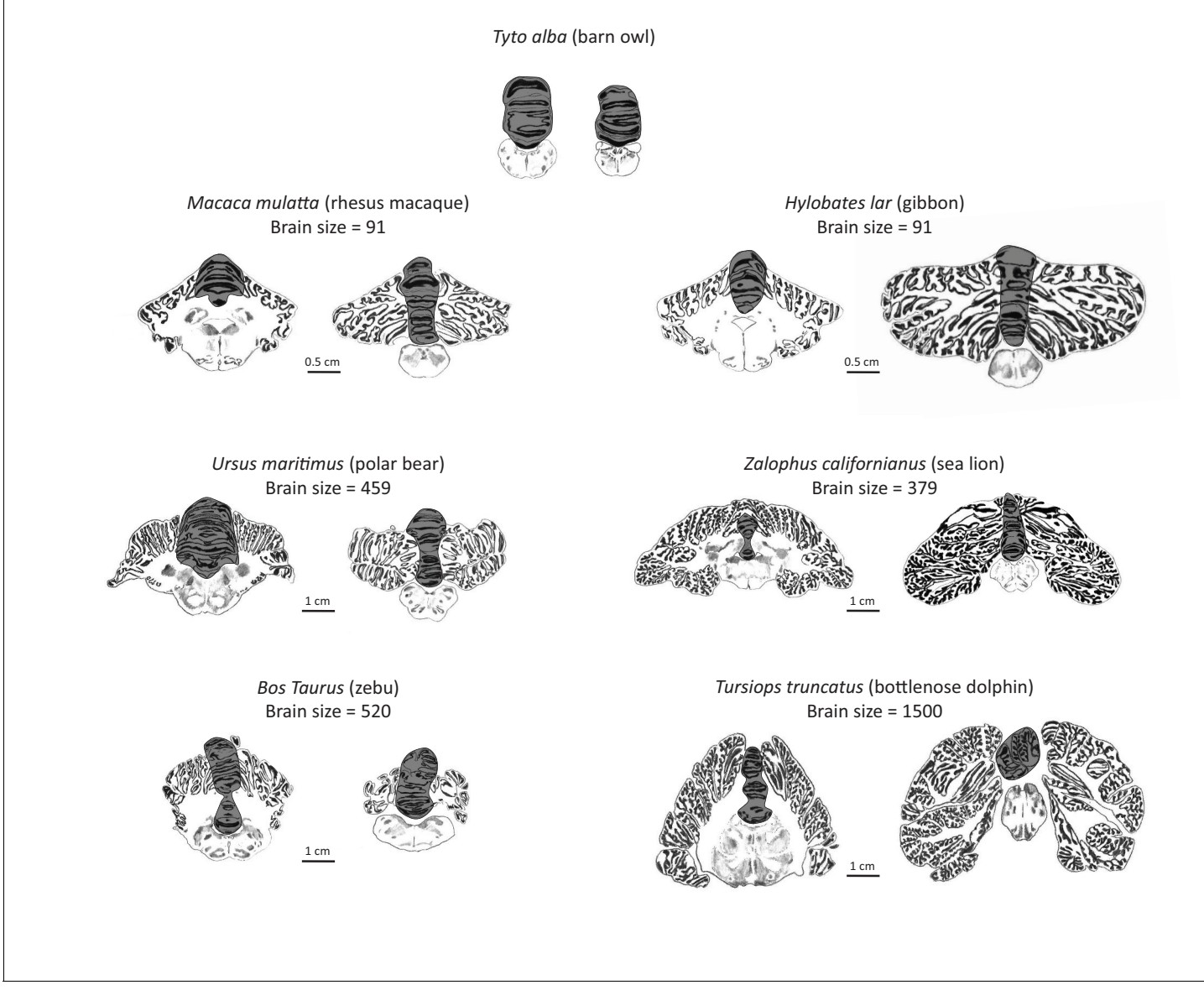

**Figure 1.** Artist's rendering of scans of brain sections from representative species in the sample. For each species, the left pane depicts a coronal section of the anterior cerebellum (near the facial colliculum of the rhomboid fossa), and the right pane of the posterior cerebellum (first available section in which white matter is no longer present). The dark overlay indicates the medial cerebellum. Note the absence of lateral cerebellar hemispheres in the barn owl. All illustrations are to scale, except for those of the barn owl. Original illustrations of *Macaca mulatta*, *Ursus maritimus*, *Zalophus californianus*, *Bos Taurus*, and *Tursiops truncatus* come from www.brainmuseum.org, *Tyto alba* comes from brainmaps.org, and *Hylobates lar* comes from the collection at the Vogt Institute for Brain Research (*Zilles et al., 2011*).
DOI: https://doi.org/10.7554/eLife.35696.003

focus on the relative measure of lateral to medial cerebellar volume to account for the functional, connectional, and developmental modularity of the cerebellum (see more details in Materials and Methods). Volumetric measurements of cerebellar partitions were used because cerebellar volume is a nearly linear function of its number of neurons (*Herculano-Houzel, 2010*), and by extension, its relative investment in particular information processing loops (*Herculano-Houzel, 2010*).

## Results

Phylogenetic scaling of lateral to medial cerebellar volume indicates a positive scaling trend ($F$ = 294.6, p<0.001, $\lambda$ = 0.945) with a slope that is higher than unity (95% confidence interval:

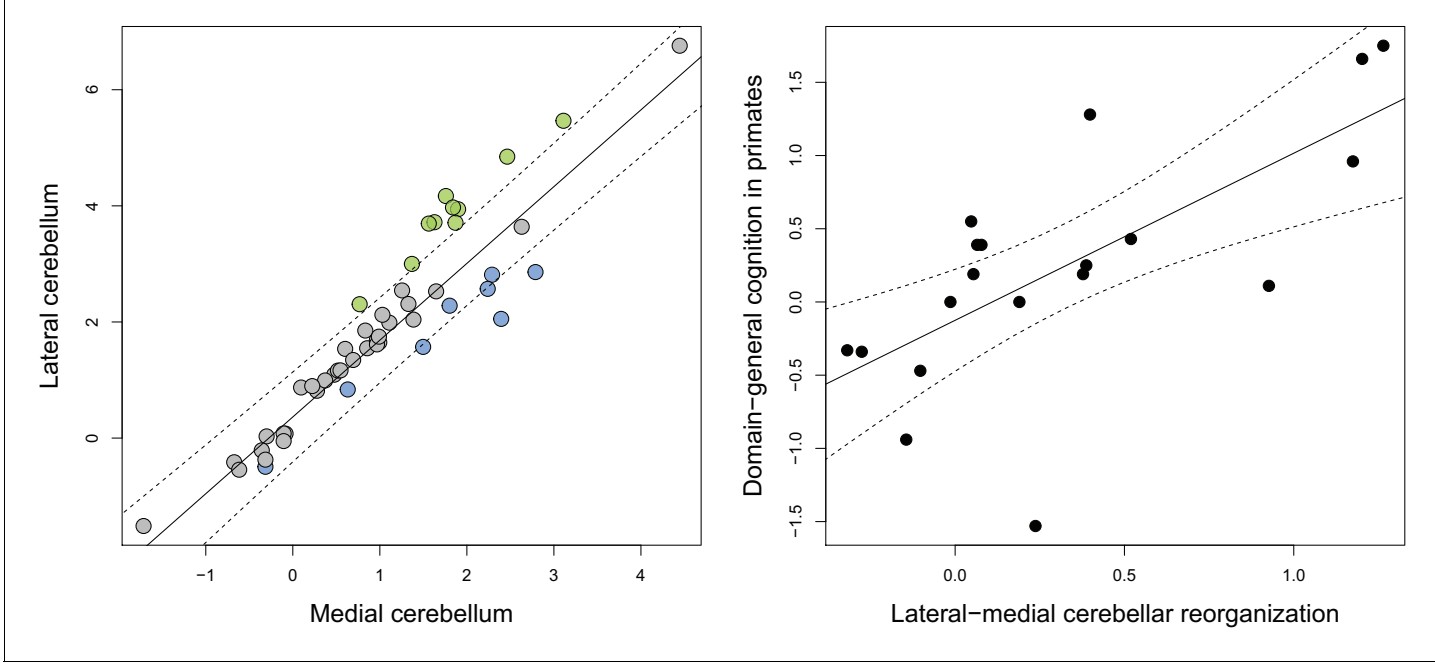

**Figure 2.** Phylogenetic generalized least-squares analysis of lateral cerebellar size to medial cerebellar size (left pane; colors as in *Figure 3*), and of lateral-medial cerebellar reorganization to a measure of domain-general cognition in primates (right pane). The original measure of domain-general cognition (*Deaner et al., 2006*) is inversely related to cognitive ability (low scores indicate high cognitive ability). Here, for the purposes of visualization, we inversed this measure so that higher scores indicate a higher cognitive ability. Phylogenetic confidence intervals were computed following Smaers and Rohlf (*Smaers and Rohlf, 2016*).

DOI: https://doi.org/10.7554/eLife.35696.004

The following source data is available for figure 2:

**Source data 1.** Brain data used in the analyses.
DOI: https://doi.org/10.7554/eLife.35696.005

1.167:1.478) (*Figure 2*). Residuals were considered as measures of 'relative lateral to medial cerebellar size' (or 'lateral-medial cerebellar reorganization'). The ratio of the observed to the predicted values range from 2.3 to 4.4 in apes, cetaceans and pinnipeds, from 0.6 to 0.7 in feliformes, and from 0.2 to 0.3 in artiodactyls. Phylogenetic regression analysis also demonstrated that lateral-medial cerebellar reorganization is a significant predictor of domain-general cognition in primates ($F$ = 15.670, p=0.001, *Figure 2*)

The evolutionary history of lateral-medial cerebellar reorganization was quantified using a Bayesian reversible-jump Ornstein-Uhlenbeck ('OU') approach (*Uyeda and Harmon, 2014*) (*Figure 3*). This analysis indicates five shifts in mean value with a posterior probability ('*PP*')>0.8. These regime shifts occurred at the root branches of the apes, the cetartiodactyls, the cetaceans (note that our sample includes toothed whales only), the pinnipeds, and the feliformes (*Figure 3*, *Figure 3—figure supplement 1*). The signal-to-noise ratio of this estimated pattern is 52.34, demonstrating that the analysis has high effect size and high power. Phylogenetic analysis of covariance (*Smaers and Rohlf, 2016*) indicates that these shifts represent significant differences in the intercept of lateral to medial cerebellar scaling (i.e. grade shifts; *Table 1*). Specifically, apes, toothed whales and pinnipeds are not significantly different from each other, but each (and as a group) are significantly different from others. Furthermore, feliformes and artiodactyls are not significantly different from each other, but each (and as a group) are significantly different from others. These results demonstrate that the six regimes identified by OU modelling constitute three significantly different grades (in order of magnitude of relative lateral to medial cerebellar size): apes, toothed whales, and pinnipeds (grade 1); rest of the sample (grade 2); artiodactyls and feliformes (grade 3). The equality of slopes assumption of analysis of covariance is upheld (grade 1 versus grade 2: $F$ = 0.009, p=0.925; grade 1 versus grade 3: $F$ = 0.088, p=0.771; grade 2 versus grade 3: $F$ = 0.836, p=0.367).

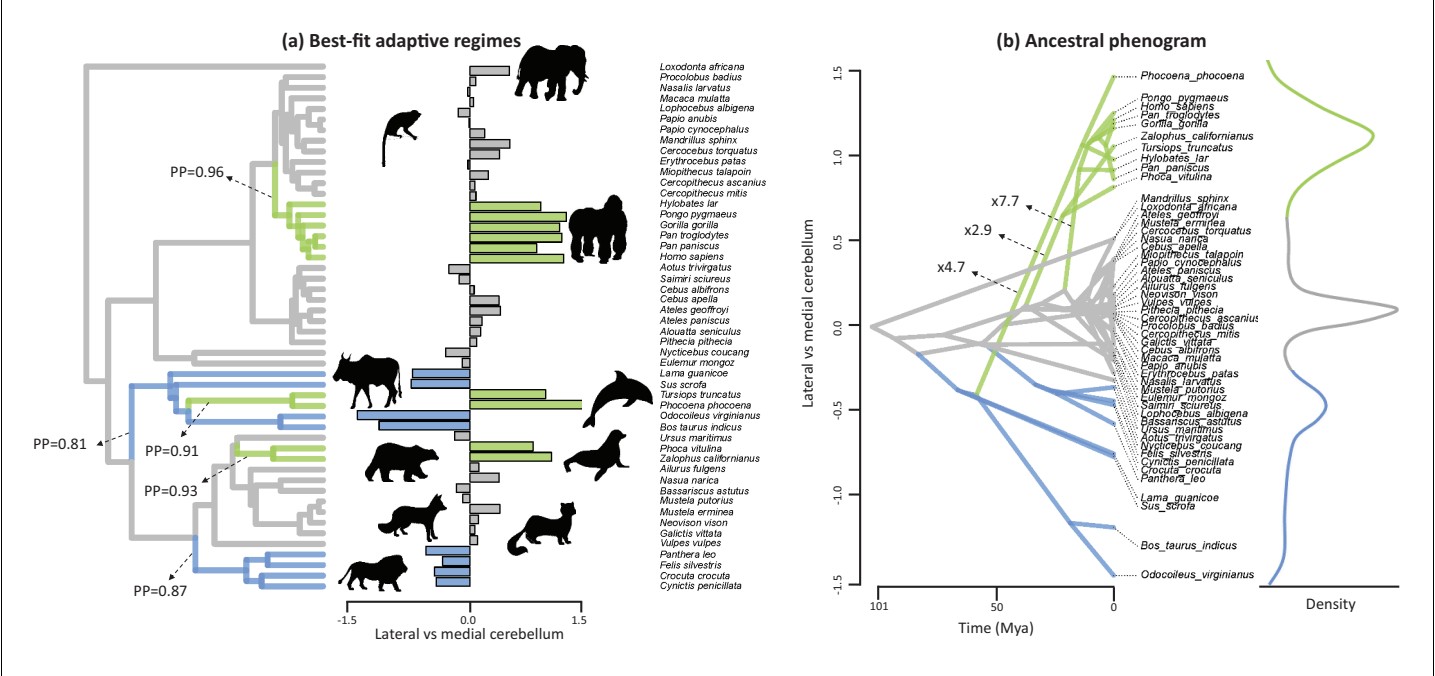

**Figure 3.** Best-fit adaptive regimes and ancestral phenogram for lateral-medial cerebellar reorganization. Best-fit adaptive regimes were estimated using a Bayesian reversible-jump procedure for fitting OU models (*Uyeda and Harmon, 2014*) and confirmed as significant grade shifts using a phylogenetic ANCOVA (*Smaers and Rohlf, 2016*). Posterior probabilities (PP) of regime shifts were estimated using the Bayesian reversible-jump procedure. Nodal values for the ancestral phenogram were estimated using a multiple variance Brownian motion approach (*Smaers et al., 2016*). Green data points and branches comprise the convergent regimes of apes, toothed whales and pinnipeds, blue data points and branches comprises those of feliformes and artiodactyls.

DOI: https://doi.org/10.7554/eLife.35696.006

The following source data and figure supplements are available for figure 3:

**Source data 1.** Tree used in the analyses.

DOI: https://doi.org/10.7554/eLife.35696.010

**Figure supplement 1.** Results of the Bayesian reversible-jump Ornstein-Uhlenbeck approach for lateral-medial cerebellar reorganization, relative cerebellum size, medial cerebellum size, and lateral cerebellum size.

DOI: https://doi.org/10.7554/eLife.35696.007

**Figure supplement 2.** Ancestral phenograms using a reversible-jump ('rjBM'), multiple variance ('mvBM') and standard Brownian motion (or 'constant variance' Brownian motion; 'cvBM') approach.

DOI: https://doi.org/10.7554/eLife.35696.008

**Figure supplement 3.** Lineage-specific rates of evolution relative to a gradual model using the mvBM and rjBM approaches.

DOI: https://doi.org/10.7554/eLife.35696.009

The evolutionary history of lateral-medial cerebellar reorganization was also examined by visualizing the evolutionary trait space in an ancestral phenogram (*Figure 3*). Ancestral states were inferred using a multiple variance Brownian motion ('mvBM') approach (*Smaers et al., 2016*). Results using a standard BM and a reversible-jump BM method yielded similar results (*Figure 3—figure supplement 2*). Lineage-specific amounts of evolutionary change were also estimated using the mvBM approach and compared against a null model of gradual evolution to obtain estimates of how much faster lineages evolve relative to a gradual model of evolution. These results are visualized in *Figure 3* and presented in full in *Figure 3—figure supplement 3*. Results using a reversible-jump BM method yielded similar results (*Figure 3—figure supplement 3*).

The same procedures were used to analyze relative cerebellum size (residuals of total cerebellar volume to the volume of the rest of the brain). These analyses revealed no regime shifts indicating PP >0.8, and only a single regime shift with PP >0.5 (the ancestral lineage of the zebu (*Bos taurus indicus*): PP = 0.76). Two other regimes shifts had 0.2 < PP < 0.5: the root of musteline carnivorans (0.38), and cercopithecine primates (0.29) (*Figure 3—figure supplement 1*). For relative cerebellar

**Table 1.** Results from a phylogenetic analysis of covariance (*Smaers and Rohlf, 2016*).
Results relate to tests of differences in intercept among groups with the slope held constant. 'Others' refers to all species in the sample not included in the other allocated groups. The analysis includes the comparison of multiple treatment groups (group a 'versus' group b) to a control group ('|' group c). High, medium, low indicates which groups have the highest, medium, and lowest trait values.

**pANCOVA**

| Group allocation | df | F | P | |
|---|---|---|---|---|
| *Convergence among regimes* | | | | |
| Apes *versus* toothed whales, pinnipeds \| others | 1,46 | 0.810 | 0.373 | *Ns* |
| Toothed whales *versus* apes, pinnipeds \| others | 1,46 | 3.488 | 0.068 | *Ns* |
| Pinnipeds *versus* apes, toothed whales \| others | 1,46 | 0.195 | 0.661 | *Ns* |
| Artiodactyls *versus* feliformes \| others | 1,46 | 1.911 | 0.174 | *Ns* |
| *3 grade model* | | | | |
| Apes, toothed whales, pinnipeds *versus* others *versus* artiodactyls, feliformes | 2,46 | 28.819 | <0.001 | *** |
| Apes, toothed whales, pinnipeds *versus* others \| artiodactyls, feliformes | 1,46 | 35.980 | <0.001 | *** |
| Artiodactyls, feliformes *versus* others \| apes, toothed whales, pinnipeds | 1,46 | 8.374 | 0.006 | ** |

DOI: https://doi.org/10.7554/eLife.35696.011

size observed/predicted values ranges from 1.2 to 1.6 for musteline carnivorans, from 0.7 to 0.9 for cercopithecine primates, and 0.5 for the zebu. Phylogenetic analysis of covariance ('pANCOVA') confirms that these regimes form significantly different grades (*Supplementary file 1*). Phylogenetic regression analysis demonstrated that relative cerebellum size is not a significant predictor of domain-general cognition in primates (lateral-medial: $F$ = 2.122, p=0.163).

The difference in rate of evolution between lateral-medial cerebellar reorganization and relative cerebellum size was tested using a Q-mode approach (*Adams, 2014*). Rates were found to be significantly different (p=0.002) at a ratio of 2.9 (lateral to medial cerebellum $\sigma^2$ = 0.00753, relative cerebellum size $\sigma^2$ = 0.00261). *Figure 4* visualizes this rate difference using a standard BM MCMC procedure (*Revell, 2012*). Similar results were found using mvBM and rjBM procedures.

We also assessed potential grade shifts in the size of the medial cerebellum relative to the rest of the brain in order to ascertain whether the described convergent trend between apes, toothed whales and pinnipeds may be confounded by different patterns of evolution in the medial cerebellum (e.g. the medial may be exceptionally small in some clades but not others, confounding the lateral to medial comparison). Results indicate that apes, toothed whales and pinnipeds are not significantly different in relative medial cerebellum size (apes versus toothed whales and pinnipeds: $F$ = 1.06, p=0.31; toothed whales versus apes and pinnipeds: $F$ = 1.11, p=0.30; pinnipeds versus apes and toothed whales: $F$ = 0.07, p=0.79). Furthermore, analysis of lateral cerebellum size versus rest of brain size yields similar results as the lateral to medial comparison in that apes, toothed whales and pinnipeds are not significantly different from each other, but are different from other mammals (apes versus toothed whales and pinnipeds: $F$ < 0.01, p=0.96; toothed whales versus apes and pinnipeds: $F$ = 1.13, p=0.29; pinnipeds versus apes and toothed whales: $F$ = 0.56, p=0.45; apes, toothed whales and pinnipeds versus others (holding constant artiodactyls and feliformes): $F$ = 5.06, p=0.01). These results confirm that the convergent lateral to medial reorganization among these clades is not due to a differential effect on medial and/or lateral cerebellum size, but rather, that it is due to a similarly convergent pattern of lateral to medial reorganization.

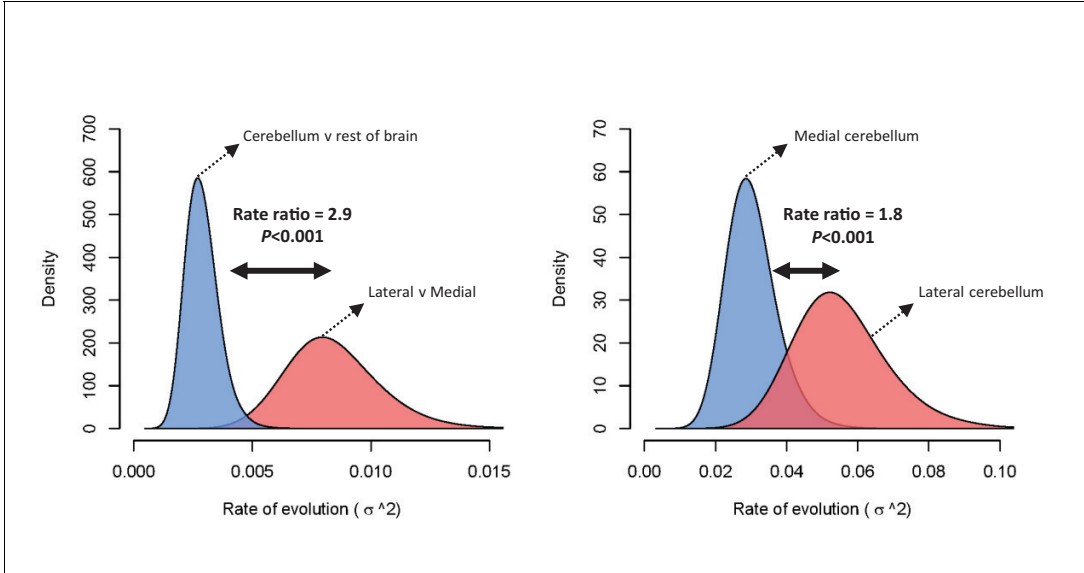

**Figure 4.** Rate of evolution ($\sigma^2$) for relative cerebellum size and lateral-medial cerebellar reorganization (left panel), and medial and lateral cerebellar volume (right panel) as estimated by a standard Brownian motion model (**Revell, 2012**). Rate ratios and *P* values are calculated using Q-mode rate analysis (**Adams, 2014**). Reversible-jump (**Venditti et al., 2011**) and multiple variance (**Smaers et al., 2016**) Brownian motion models yield equivalent results.

DOI: https://doi.org/10.7554/eLife.35696.012

## Discussion

Results demonstrate a combination of parallel and convergent evolutionary events in mammals leading to relative lateral cerebellar expansion. This evolutionary pattern encompasses distantly related lineages - apes, toothed whales, and pinnipeds. The congruence between apes and pinnipeds demonstrates parallel evolution as they derive from a similar ancestral condition. The congruence with toothed whales demonstrates convergence as they derive from a different ancestral condition (evident in the artiodactyls grade with smaller relative lateral cerebellar size). These results bear on the interpretation of the evolution of cognition, modularity in brain evolution, and the occurrence of neurodevelopmental shifts across species.

The cerebellum has been hypothesized to be involved in the domain-general automation of associative learning abilities (**Schmahmann, 1997**) and, according to the results of the present analyses, may be a target of evolution leading to the emergence of 'intelligent' behaviors. Specifically, the contribution of the cerebellum's role to cognition has been hypothesized to lie in the extrapolation of error-based motor learning processes to domains beyond motor control such as social cognition (**Sokolov et al., 2017**). According to theoretical models of cerebellar function, the cerebellum receives a copy of the motor command and generates a representation ('internal model') of the expected consequences of that command (**Sokolov et al., 2017**; **Ito, 2008**; **Moberget and Ivry, 2016**). Such sensory prediction ('forward model') allows for predictive control of motor actions which results in smooth and automated execution (**Koziol et al., 20132014**). In humans, evidence has been found that this process, in addition to being active in relation to motor actions, may also be active in semantic processing (**Moberget et al., 2014**), and social cognition (**Van Overwalle et al., 2014**). Here, we demonstrate a significant relationship between lateral-medial cerebellar reorganization and a measure of domain-general cognitive capacity across primates (**Figure 2**). This measure of cognitive ability in primates quantifies the performance of animals on nine tasks that are independent of perceptual bias and contextual confounds (**Deaner et al., 2006**). This positive correlation provides indirect support for the hypothesis that the cerebellum is involved in cognition, and further suggests that this association may have deep evolutionary roots.

Ideally, a comparative test between lateral cerebellar expansion and cognition would include all mammalian species. Unfortunately, Primates are the only order for which a standardized measure of higher cognitive capacity is available that matches available cerebellar data. Beyond primates, the

analysis of vocal production learning ('VPL') has, however, also produced useful insights into the evolution of higher order learning capacities. VPL is defined as the ability to modify existing vocalizations and to imitate novel sounds not belonging to the innate repertoire (*Janik and Slater, 2000*). VPL aligns with 'cerebellar-type' error-based learning in that it results in smooth and automated execution after combining individual motor actions into a more complex automated model. In humans, the link between cerebellum and vocal learning is supported by an association between lateral cerebellar activation patterns and semantic processing (*Moberget et al., 2014*). In mammals, only a few species are known to be capable of VPL: humans, bats (*Vernes, 2017*), elephants (*Stoeger et al., 2012*), seals (*Ralls et al., 1985*; *Ravignani et al., 2016*), dolphins (*Reiss and McCowan, 1993*), and toothed whales (*Foote et al., 2006*). Apart from bats (not included in our sample due to a lack of available data on cerebellar hemispheres), our analyses indicate that all these species have increased lateral cerebella (though note that although our analysis demonstrates that elephants indicate a pronounced trend of lateral cerebellar expansion, this trend is not significant in our sample). The congruence between our results on multiple independent evolutionary expansion events of the mammalian lateral cerebellum and the occurrence of the capacity for VPL is further support of an evolutionary association between changes in cerebellar processing and cognition.

A potential link between such cognitive specializations and convergent lateral cerebellar expansion suggests that repeated selection on a common neural substrate has, at least partly, characterized mammalian brain evolution in relation to cognition. This, however, does not exclude the concomitant occurrence of lineage-specific adaptive specializations in other parts of the brain, such as the neocortex. The available information suggests that different orders of mammals may have neocortices that are organized in fundamentally different ways. Primates, for example, have evolved a suite of novel fronto-parietal cortical areas that form a network that is functionally, hodologically, and developmentally linked. Crucially, this primate-specific network is not observed in other mammals (*Preuss, 2007*; *Preuss, 1995*; *Preuss and Goldman-Rakic, 1991a*; *Preuss and Goldman-Rakic, 1991b*). Such order-specific specializations lend support to the hypothesis that species differences in intelligence can, at least partly, be understood as an interaction between aggregates of domain-specific abilities, together with changes in domain-general processing. Indeed, our current results demonstrate that those cerebellar partitions that are hypothesized to underpin domain-general associative learning abilities have repeatedly been subject to directional selection, suggesting that modification of common neural systems may contribute significantly to the evolution of animal intelligence. The nature of cerebellar microstructural circuit anatomy (fairly uniform across mammals) and hypothesized function (to automate and streamline cerebral information processing) lends support to our conclusion that its domain-general features have been repeatedly subject to selection, irrespective of the precise organization of the cerebral cortex in different orders. Across orders, the evolution of the mammalian brain may hereby be characterized by something new (cerebral specializations) and something borrowed (cerebellar convergences).

The potential link between lateral cerebellar expansion and cognition across species, may further inform on the nature of the cognitive differences across species. In primates, the lateral cerebellum's role in cognitive processing is tightly linked to higher-order visual and somatosensory cortical association areas (*Preuss, 2007*). Primates evolved specializations for visual grasping and reaching that involved extensive integration of visual, motor, and somatosensory processing (*Wise et al., 1997*; *Felleman and Van Essen, 1991*). Accordingly, cortical association areas involved in these specializations are connected to lateral cerebellar lobules (*O'Reilly et al., 2010*; *Ramnani et al., 2006*; *Glickstein et al., 2011*; *Sokolov et al., 2014*; *Schmahmann and Pandya, 1991*). The nature of cortico-cerebellar hodology in toothed whales and pinnipeds is, however, less well described. Although previous research provides indirect evidence for the expansion of cortical association areas in these groups (*van Kann et al., 2017*; *Hof and Van Der Gucht, 2007*; *Hof et al., 2005*; *Turner et al., 2017*; *Sawyer et al., 2016*), more research is needed. It is clear, however, that cortical information processing in toothed whales is geared more towards auditory processing (echolocation), while in pinnipeds, as in terrestrial carnivorans and primates, it is geared more towards vision. Rather than evolving echolocation to navigate a poorly visible aquatic niche, pinnipeds evolved exceptional somatosensory and visual sensitivity (*Hanke et al., 2009*; *Scholtyssek et al., 2008*; *Dehnhardt et al., 1998*). The differences between toothed whales (auditory specializations for echolocation) and primates and pinnipeds (visual and somatosensory specializations) accord with preliminary observations that the enlargement of the lateral cerebellum in toothed whales may be due to

the expansion of different lobules within the hemisphere, as compared with apes and pinnipeds. Whereas hemispheric expansion in apes is most likely due to the enlargement of lobule HVII (*Balsters et al., 2010*) (which connects closely to cortical association areas such as temporal, parietal and prefrontal cortex, which are involved in visually-guided actions), hemispheric expansion in toothed whales may be due to enlargement of lobule HIX (*Jansen and Jansen, 1969*) (possibly more closely connected to auditory processing). Given that visual and somatosensory specializations are prominent in both primates and pinnipeds, we speculate that hemispheric expansion in pinnipeds is also due to lobule HVII. Similarly, visually dominant birds such as crows, parrots, and woodpeckers have enlarged visually projecting lobules VI–IX (in the medial cerebellum; birds do not have lateral extensions of the cerebellum, see Figure 1), likely related to their repertoire of visually guided goal-directed beak behavior (*Sultan, 2005*).

The lower than expected relative lateral cerebellar size in artiodactyls and felines should be confirmed with increased sampling. Considering that the lateral expansion of cerebellar lobules is a feature that first arose in early mammals, the artiodactyls and felines may, in fact, better reflect an early ancestral condition. In this scenario, the extent of the convergent and parallel expansion in apes, toothed whales, and pinnipeds observed here would be an underestimation of the true trend.

Previous studies on cerebellar evolution have mainly considered the cerebellum as a whole. These studies have suggested that the cerebellum underwent rapid size increase (relative to the cerebral cortex considered as a whole) in apes (*Barton and Venditti, 2014*), and that the elephant has the largest relative cerebellum size of all mammals (*Maseko et al., 2012*). After accounting for scaling differences with the rest of brain size, our results confirm that relative cerebellum size shows a grade shift in apes (pANCOVA, $F$ = 5.432, p=0.024), but we found no evidence to support the conclusion that relative cerebellar size is significantly expanded in elephants compared to other mammals (pANCOVA, $F$ = 2.426, p=0.126). However, Bayesian reversible-jump estimation of multi-peak OU models indicates that neither of these patterns are the most dominant in characterizing the macroevolutionary landscape of relative cerebellum size. Rather, the macroevolutionary landscape of relative cerebellar size is primarily characterized by a significant decrease in the zebu (observed/allometrically predicted value = 0.52) and cercopithecine primates (observed/allometrically predicted value = 0.85), and an increase in the musteline carnivorans (observed/allometrically predicted value = 1.44) (SI1, SI5). Furthermore, a comparison of the rate of evolution of lateral-medial cerebellar reorganization against relative cerebellum size shows a significantly higher rate for lateral-medial reorganization (*Figure 4*), suggesting that the main result of selective pressure in mammalian cerebellar evolution has led to modular changes within the cerebellum (likely related to reciprocal loops of information processing with particular cerebral networks), rather than global changes in cerebellum size relative to overall brain size or overall cerebral cortex (or 'neocortex') size. These results also accord with the cerebellum's functional, connectional, and developmental modularity.

The interpretation of the nature of structural volumetric reorganization in brain evolution has also been debated. Some argue that volumetric reorganization occurs predominantly in a 'mosaic' fashion as a result of behavioral selective pressures which is largely unconstrained by developmental patterning (*Barton, 2006*; *Barton and Harvey, 2000*; *Barton, 2001*). Others argue that the sequence of developmental events plays a more significant role in shaping brain reorganization such that earlier developing structures are more likely to be evolutionarily conservative than later developing structures (*Finlay et al., 2001*; *Finlay and Darlington, 1995*). Our results indicate that cerebellar reorganization occurs in alignment with developmental predictions, with the later developing partition (i.c. the lateral cerebellum) showing a higher rate of evolution and characterized by a more complex evolutionary scenario than the earlier developing partition (i.c. the medial cerebellum) (*Figure 3—figure supplement 1*, *Figure 4*). Mammalian cerebellar reorganization is therefore more consistent with modular change (i.e. reorganization predicated on developmental patterning), than it is with mosaic change (i.e. reorganization largely unconstrained by developmental patterning). In primates, lateral cerebellar evolution appears to be particularly linked with those cerebral association areas that are part of the primate-specific cerebro-cerebellar network (*Smaers, 2014a*; *Balsters et al., 2010*; *Smaers et al., 2013*; *Smaers et al., 2011*), where a similar grade shift in great apes and humans is observed (*Smaers et al., 2017*; *Passingham and Smaers, 2014*).

The occurrence of significant changes in cerebellar structural reorganization in the mammalian macroevolutionary landscape (*Figure 3*), combined with the higher rate of evolution for changes in lateral-medial cerebellar organization over changes in relative cerebellar size (*Figure 4*) draws

attention to a new avenue of adaptive brain evolution. Across mammals, the total number of cerebellar neurons (excluding Purkinje neurons) correlates significantly with the number of cerebral neurons, leading to the suggestion that coordinated processing networks with the neocortex constrain cerebellar evolution(*Herculano-Houzel, 2010*) To date, however, such analyses have rarely accounted for modular reorganization within gross anatomical structures (such as the cerebellum and the neocortex) independently of overall size (but see *Balsters et al. (2010)*). The significant grade shifts in cerebellar reorganization observed here show that not all scaled up cerebella are anatomically homologous. A comparison of observed to allometrically predicted values indicate that apes, toothed whales, and pinnipeds have lateral cerebellar hemispheres that are 2.3 to 4.4 times larger than predicted relative to the medial cerebellum, while artiodactyls have lateral cerebella that are 3.3 to 4.4 times smaller than predicted. Given near isometric scaling of number of neurons and mass in the cerebellum, this implies that the relative number of neurons dedicated to automating either higher cerebral association processing (lateral cerebellum), or basic motor skills and proprioception (medial cerebellum) is similarly unevenly distributed in apes, toothed whales, and pinnipeds versus artiodactyls. Moreover, the artiodactyls in our sample have similarly sized brains than apes and pinnipeds, further demonstrating the importance of modular reorganization patterns that are independent of brain size. Other aspects of cerebellar microstructural anatomy may also exhibit functionally significant phylogenetic variation. For instance, cerebellar cells (mostly granule cells) are more densely packed in eulipotyphlans, primates, and elephants, compared to other mammals investigated (*Herculano-Houzel et al., 2015*). And remarkable differences in the ratio of granule cells to Purkinje neurons have been reported, with the greatest proportions of granule cells per Purkinje neuron found in primates, toothed whales, and elephants (*Lange, 1975*). Additionally, Golgi impregnation studies have demonstrated that cerebellar neuron morphologies vary across mammals, showing strikingly extensive dendritic branching of Lugaro cells in elephants (*Jacobs et al., 2014*). The functional impact of such species differences in microstructure is yet to be fully understood, but should be considered alongside volumetric reorganization in a comprehensive model of cerebellar evolution.

## Conclusions

Further work is needed to expand the detail of the cerebellar delineations, the breadth of the comparative neuroanatomical sample, and the range of behavioral measures on associative learning abilities across mammals. Expanding the detail of cerebellar delineations would allow evaluating the extent to which the currently observed macroevolutionary pattern of convergence towards lateral-medial cerebellar reorganization may be driven by different patterns of modularity within the lateral cerebellum across different clades (e.g. lobule HVII in apes and pinnipeds, lobule HIX in toothed whales). Further expanding the breadth of the comparative sample and the detail of neurobiological measurements will allow increasing the resolution of evolutionary inference, expanding our understanding of neurobiological modification in relation to different body plans and life styles, and consequently, refining our understanding of the evolutionary pathways that have shaped intelligent behavior in vertebrates. Some outstanding questions on species that are not covered by our current sample include the putative expansion of lateral cerebellar hemisphere in bats (in relation to echolocation and the expansion of the paraflocculus [*Larsell, 1970*; *Larsell and Dow, 1935*]), and potential differences in lateral cerebellar expansion in baleen versus toothed whales (baleen whales do not echolocate and may therefore not indicate an expansion of lobule HIX, as observed in toothed whales [*Jansen and Jansen, 1969*]).

We conclude that a tendency for distantly related mammalian species to converge on lateral-medial cerebellar reorganization plays an important role in explaining cerebellar macroevolution. Considering the lateral cerebellum's hypothesized role in automating higher-order cortical association information processing, this macroevolutionary pattern suggests a tendency for distantly related species to independently acquire 'cerebellar-like' associative-learning abilities. We propose cerebellar reorganization as a target for broad comparative investigations of neurobiological diversification because it is more reflective of modularity and interconnectivity than overall brain size, and more validly represents homologous functional and neural circuitry than the more traditional focus on overall neocortex.

## Materials and methods

### Data

Brain data were taken from *MacLeod et al. (2003)*, *Smaers et al. (2011)*, and *Maseko et al. (2012)*. For the anthropoid data, preference was given to data presented in *MacLeod et al. (2003)* because it includes more individuals per species. Data for anthropoid species not presented in *MacLeod et al. (2003)* were then taken from *Smaers et al. (2011)*. *Smaers et al. (2011)* used the same delineation protocol as *MacLeod et al. (2003)*, and also used brains processed in the same lab (*Zilles et al., (2011)*). *Maseko et al. (2012)* collected additional data using both histological sections (using similar delineation criteria as *MacLeod et al. 2003*] and MRI images (for the elephant and harbor porpoise only). The comparability between MRI and histological data likely involves a degree of error, although this error was suggested by *Maseko et al. (2012)* to be minimal. Data are presented in *Figure 2—source data 1*. Behavioral data were taken from *Deaner et al. (2006)*. Other data sets of domain-general cognition were considered (*Benson-Amram et al., 2016*; *MacLean et al., 2014*), but found to have a limited overlap with the available neuroanatomical data (≤7 species).

### Phylogeny

The phylogeny was adjusted from *Faurby and Svenning (2015)*, who used a novel heuristic-hierarchical Bayesian approach for estimating a species-rich (>4100 species) phylogeny of mammals. In their approach, species with a large amount of sequence data are freely placed in a standard Bayesian MCMC procedure. The phylogenetic placements of species with decreasing data quantities are estimated with increasing restrictions on their possible placement. Finally, species with no sequence data are placed based on morphological trees or existing taxonomy. Additional details can be found in the authors' full description of their procedure. The result of their procedure is a sample of 1000 trees from the final posterior distribution. We chose to use the 4160 species tree as this represents the largest possible tree of species all with unambiguous placement in the phylogeny. Faurby and Svenning estimated branch lengths on these final trees using a two-step process where some higher-level divergences were manually incorporated from other sources and the remaining branch lengths simulated using the age of the clade and either a Yule or Birth-Death model of evolution. Our analysis required a single resolved tree. A typical consensus of the 1000 sampled trees would result in negative branch lengths. We instead used the maximum clade credibility tree (MCC) from the sample, as estimated using TreeAnnotator v2.3.1 (*Drummond et al., 2012*). The resulting tree is presented in *Figure 3—source data 1*. For the purposes of our analyses, this tree was pruned to contain only those species in our sample.

### Measure of relative size

To evaluate whether a particular brain structure is enlarged relative to other structures (or the rest of the brain), the standard approach has been to fit a (phylogenetic) regression line through a comparative sample and to calculate to what extent predicted values correspond to observed values (*Passingham, 1973*). The focus of our study lies on the comparison between the lateral and medial cerebellum. This measure quantifies changes within the cerebellum between its two major constituent partitions that are functionally, connectionally, and developmentally distinct. Whereas the medial (vermis and paravermis) cerebellum is involved in basic motor control, proprioception and autonomic functions, the lateral hemispheres are the site of integration for multiple streams of cerebral information processing (*Glickstein et al., 2011*). We also ran analyses using an alternative measure that considers the comparison of overall cerebellar size relative to the size of the rest of the brain. Although this latter measure is the most commonly used in previous research, it overlooks modularity within the cerebellum. Moreover, this measure also does not account for the fact that the cerebellum is highly interconnected with much of the rest of the brain. A comparison against the rest of the brain thus performs a statistical control for much of what is neurobiologically relevant. We primarily focus on the measure of lateral to medial cerebellum because it represents the cerebellum's modular organization and is therefore more relevant to understanding the underpinnings of neural information processing (*Passingham and Smaers, 2014*).

## Evolutionary modelling

To identify the evolutionary dynamics of brain region enlargement we utilize phylogenetic comparative methods that reveal the tempo, mode, and history of trait evolution. Using a phylogenetic tree and observed information from contemporary tip taxa, these methods employ statistical and mathematical models of evolution to describe the pattern and rate of trait change along individual branches of a phylogeny. As such, these methods infer the temporal origin and rate of evolution of a trait across a phylogenetic landscape.

The most frequently used statistical model of evolution is standard Brownian motion ('BM'), which assumes that traits change at each unit of time with a mean change of zero and unknown and constant variance (*Cavalli-Sforza and Edwards, 1967*; *Felsenstein, 1973*; *Felsenstein, 1985*). Within Brownian motion, the evolution of a continuous trait 'X' along a branch over time increment 't' is quantified as $dX(t) = \sigma dB(t)$, where '$\sigma$' constitutes the magnitude of undirected, stochastic evolution ('$\sigma2$' is generally presented as the Brownian rate parameter) and 'dB(t)' is Gaussian white noise. The standard BM model of evolution is ideally suited not only as a baseline model for hypothesis testing approaches such as least-squares analysis (ANOVA, ANCOVA, GLS), but also as a baseline model for rate analysis.

The standard BM model is, however, less well suited for estimating the evolutionary history of biological traits as it assumes that the rate of evolution is constant across the sample. Therefore, approaches that aim to model the evolutionary history of biological traits commonly incorporate additional parameters to capture possible deviations from the standard gradual mode of evolution assumed by BM. Ornstein-Uhlenbeck ('OU') models incorporate stabilizing selection as a constraint and hereby quantify the evolution of a continuous trait 'X' as $dX(t) = \alpha[\theta - X(t)]dt + \sigma dB(t)$ where '$\sigma$' captures the stochastic evolution of BM, '$\alpha$' determines the rate of adaptive evolution towards an optimum trait value '$\theta$' (*Hansen, 1997*). This standard OU model has been modified into multiple-regime OU models allowing optima to vary across the phylogeny (*Butler and King, 2004*). Such multi-regime OU models allow modelling trait evolution towards different 'regimes' that each display a different mean trait value. Several methods have been developed that use this modelling approach to model trait diversification by estimating shifts in $\theta$-values along the branches of the phylogeny (e.g., *Uyeda and Harmon, 2014*; *Khabbazian et al., 2016*). Multi-rate BM approaches expand the standard BM model by including additional rate parameters that capture potential differences in rates among different clades or lineages. Venditti et al (*Venditti et al., 2011*; *Pagel and Meade, 2013*) use a reversible-jump algorithm ('rjBM') in a Bayesian MCMC framework to estimate where such potential shifts in rate may have occurred. *Smaers et al. (2016)* use a heuristic algorithm ('mvBM': multiple variance BM) that leverages global and local information to estimate rates of evolution for each lineage in the tree.

It is clear that these different approaches have different strengths and weaknesses, and should therefore be used within the constraints of what they aim to do. OU modelling approaches, for example, are commonly agreed to be a very powerful approach for modelling trait diversification, though recent research has pointed towards some challenges when using such models. Specifically, the theoretical properties of the maximum-likelihood estimators for OU parameters can result in non-uniqueness and inaccuracy causing traditional model selection criteria to favor overly complex scenarios (*Lst and Ané, 2014*). More recent Bayesian (*Uyeda and Harmon, 2014*) and least-squares (*Khabbazian et al., 2016*) procedures, however, have proposed adjustments to traditional procedures that overcome these difficulties. Also multi-rate BM models have clear limitations. Such approaches are commonly highly parameterized (*Lst and Ané, 2014*) and therefore less suitable for hypothesis testing (*Smaers and Mongle, 2017*). Such models are, however, particularly useful for providing best-fit estimates of evolutionary history (*Smaers and Mongle, 2017*).

## Estimating changes in mean value

We modeled changes in mean values along individual branches of the phylogeny using a Bayesian reversible-jump OU procedure (*Uyeda and Harmon, 2014*; *Uyeda and Eastman, 2014*). This procedure estimates a best-fit adaptive regime configuration of cerebellar reorganization (more info in SI), whereby 'regimes' are defined as a group of lineages with a similar mean value ($\theta$ in the OU model framework). By using a Bayesian parameter estimation procedure this approach avoids the non-uniqueness of parameter estimation inherent to maximum likelihood procedures (*Lst and Ané,*

*2014*). To avoid overfitting this procedure uses a conditional Poisson distribution as a prior on the number of shifts (ranging from zero to half the number of tips). Furthermore, this procedure allows the posterior probability ('*PP*') threshold to call a shift to be adjusted so as to provide more liberal (*PP* $\geq$0.2) or more conservative (*PP* $\geq$0.8) estimations. A more liberal *PP* threshold hereby tends to result in high recall rates (many of the true shifts are detected) and low precision (many false positives are detected), while a more conservative *PP* threshold tends to result in low recall rates (some true shifts are not detected) and high precision (few false positives are detected).

## Estimating ancestral values

Ancestral values were inferred using a multiple variance BM ('mvBM') approach (*Smaers et al., 2016*; *Smaers and Mongle, 2018*). Code to implement mvBM and phylogenetic ANCOVA is available from the 'evomap' R package (*Smaers and Mongle, 2018*; copy archived at https://github.com/elifesciences-publications/evomap). This procedure provides an estimate of evolutionary history that is based on lineage-specific rates of evolution (visualized in the ancestral phenogram *Figure 3b*). This approach has been shown to provide estimates equivalent to standard BM when the trait evolves according to that model, and to outperform it when the trait does not adhere to standard BM by improving the estimation of trait evolution in those location where the evolutionary process deviates from standard BM (*Smaers et al., 2016*; *Smaers and Mongle, 2017*). In *Figure 3—figure supplement 2* and *3* we also report results obtained using a reversible-jump BM ('rjBM') method, which is a different multi-rate BM approach (*Venditti et al., 2011*; *Pagel and Meade, 2013*). This different approach provides equivalent results for the analyses presented here. Both these methods were used in a Bayesian MCMC framework using 10 million iterations and sampling every 100th iteration, which rendered normal distribution of log likelihood values for all analyses.

## Estimating lineage-specific rates of evolution

Lineage-specific variation was compared to a baseline expectation given a standard BM model to provide estimates of how much faster evolution in a particular lineage is estimated to be relative to a gradual model. The amount of change observed at each branch (the difference between descendant and ancestral branches as inferred using the mvBM and rjBM approaches) was compared with a neutral scenario in which all the species in the phylogeny were simulated to evolve at a constant rate (*Gómez-Robles et al., 2017*). For these analyses, the original phylogeny was transformed to generations. Age at first reproduction as obtained from PanTHERIA database (*Jones et al., 2009*) was used as a proxy for generation time. When this variable was not available for a given species included in our dataset, the value corresponding to the closest species with known age at first reproduction was used. The time-based phylogeny was rescaled to generations by dividing each branch length by the generation time corresponding to their descendant species or descendant inferred node. A per-generation rate of evolution was calculated based on available data (*Martins, 1994*), and it was later used to simulate evolution over the studied phylogeny at that constant rate (*Polly, 2017*; *Polly, 2004*). Simulations were repeated 100 times for each trait and differences between descendant and ancestral values were calculated. The average of those differences for each branch were used as the neutral expectation of the amount of change that each branch would have accumulated had all the branches evolved at the same rate. The ratio between observed and simulated amounts of change per branch is lower than one for slow-evolving branches and greater than one for fast-evolving branches.

## Testing estimated changes in mean value

Because estimation of evolutionary patterns is inherently uncertain we translated the estimated model from the Bayesian reversible-jump OU procedure into a least-squares framework. Least-squares analysis allows testing whether the patterning of the extant variation suggested by the evolutionary estimation is significant. We hereby used the least-squares solution to phylogenetic analysis of covariance (pANCOVA) (*Smaers and Rohlf, 2016*) to test for differences in slopes and intercepts among the extant values of the estimated regimes. This implementation of pANCOVA includes additional indicator variables describing group membership to the standard generalized least-squares procedure ($\mathbf{y} = \mathbf{Xb} + \epsilon$) (*Smaers and Rohlf, 2016*). This procedure calculates the change associated with the clades of interest in the residual variance simultaneously with the phylogenetic regression

parameters, and hereby allows for a direct test of whether a model with multiple grades (assuming multiple groups with different mean trait values) provides a significantly better fit to the data than a model with only a single grade (assuming that no particular group indicates a significantly different mean trait value). Technical details and examples of implementation are available in *Smaers and Rohlf (2016)*. Code to implement pANCOVA is available from the 'evomap' R package (*Smaers and Mongle, 2018*). We further include the λ parameter in order to account for the degree of phylogenetic signal in the data (*Pagel, 1997*). Considering the uncertainties involved in reversible-jump and Ornstein-Uhlenbeck modelling, this step provides a crucial confirmation that the estimated results from the modelling analyses are indeed significant.

## Testing for differences in rate of evolution among traits

To test for differences in rate among different measures of cerebellar reorganization, we use the procedure proposed by Adams et al (*Adams, 2014*; *Denton and Adams, 2015*; *Adams and Otárola-Castillo, 2013*). This method uses a distance-based approach (Q-mode) to quantifying evolutionary rate. Q-mode approaches provide estimates of evolutionary rates that are numerically identical to those obtained using covariance-based implementations (R-mode). The advantage of the Q-mode approach is that it can be extended to high-dimensional data while maintaining appropriate Type I error and high statistical power for detecting differences in $\sigma^2$ (25). This approach assumes a standard BM model of evolution. Hypothesis testing is performed by comparing the observed ratio of evolutionary rates with a distribution of possible ratios obtained under the null hypothesis that there is no rate difference between traits.

## Model uncertainty, reliability, and effect size

Estimating patterns of evolution along individual lineages given comparative trait data and a phylogeny is an inherently uncertain endeavor (*Lst and Ané, 2014*). Several steps can, however, be taken to confirm the reliability of the estimated patterns (*Smaers et al., 2017*).

First, when possible results should be translated to least-squares analysis. Least-squares analysis allows for hypothesis testing and can hereby confirm or falsify the patterning of the extant variation that is suggested by evolutionary modelling. This is particularly true for bivariate allometric analyses. The phylogenetic regression ('pGLS' [*Rohlf, 2001*]) and its extensions towards more complex generalized linear models (e.g. pANCOVA [*Smaers and Rohlf, 2016*]) are the most powerful hypothesis testing approaches for comparative data. Although least-squares analysis does not allow confirming lineage-specific evolutionary patterns, it is clear that the patterning of the extant variation as suggested by evolutionary modelling analysis is expected to produce significant results when used in least-squares analysis. Because observed power is a simple function of the observed *P*-value in least-squares analysis (*Hoenig and Heisey, 2001*), tests that produce significant results can be considered to have high power.

Second, proxies of effect size can be calculated for evolutionary patterns that have been estimated using OU modelling. *Cressler et al. (2015)* demonstrated that a signal-to-noise ratio ($\sqrt{\eta}\phi$) provides a better predictor of power than sample size. This ratio compares the minimum difference in mean value among regimes (multiplied by the strength of directional change among regimes) with a measure of noise intensity. *Cressler et al. (2015)* demonstrated that when $\sqrt{\eta}\phi \gg 1$, high statistical power can be inferred. Such measures of effect size are crucial indicators of reliability and can thus be used to build confidence in the accuracy of estimated patterns.

Third, reliability of the estimated patterns can further be confirmed by testing the same hypothesis using different methods with different model assumptions. If the same result is obtained regardless off methods used or models assumed, it can be concluded that the results are reliable.

We followed these three steps to confirm the patterns estimated by the Bayesian reversible-jump OU procedure. We confirmed the statistical significance of differences in intercept among three grades using pANCOVA (*Figure 3*), demonstrated that $\sqrt{\eta}\phi \gg 1$ is true for the results presented in *Figure 3*, and that the pattern presented in *Figure 3* is confirmed using pANCOVA, mvBM ancestral and rate estimation and rjBM ancestral and rate estimation.

## Acknowledgements

JBS was supported by the Wenner Gren Foundation (Gr. 9209). CCS was supported by the James S McDonnell Foundation (220020293). We thank F James Rohlf for insightful discussions and advice, Javier Lázaro Tapia for providing the artist's rendering of brain sections in *Figure 1*, and Kate Thompson for help with other illustrations.

## Additional information

### Funding

| Funder | Grant reference number | Author |
| --- | --- | --- |
| Wenner Gren Foundation | Grant 9209 | Jeroen B Smaers |
| James S. McDonnell Foundation | 220020293 | Chet C Sherwood |

The funders had no role in study design, data collection and interpretation, or the decision to submit the work for publication.

### Author contributions

Jeroen B Smaers, Conceptualization, Formal analysis, Validation, Investigation, Visualization, Methodology, Writing—original draft, Project administration, Writing—review and editing; Alan H Turner, Aida Gómez-Robles, Formal analysis, Methodology, Writing—review and editing; Chet C Sherwood, Conceptualization, Writing—original draft, Writing—review and editing

### Author ORCIDs

Jeroen B Smaers (iD) http://orcid.org/0000-0003-1741-9839

### Decision letter and Author response

Decision letter https://doi.org/10.7554/eLife.35696.016
Author response https://doi.org/10.7554/eLife.35696.017

## Additional files

### Supplementary files

• Supplementary file 1. phylogenetic ancova procedures for relative cerebellum size, using the regimes as identified by Bayesian reversible-jump estimation of multi-optima OU models (SI1) as groups.
DOI: https://doi.org/10.7554/eLife.35696.013

• Transparent reporting form
DOI: https://doi.org/10.7554/eLife.35696.014

### Data availability

The brain data and the phylogeny that were used in the analyses are available as source data files (Figure 2—source data 1, and Figure 3—source data 1). Behavioral data for primates is available from Figure 2, Deaner RO, Van Schaik CP, Johnson V. 2006. Do some taxa have better domain-general cognition than others? A meta-analysis of nonhuman primate studies. Evolutionary Psychology 4: 149-196.

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
