## [Decision Letter]

Thank you for submitting your article "A cerebellar substrate for intelligence evolved multiple times independently in mammals" for consideration by *eLife*. Your article has been favorably evaluated by Richard Ivry (Senior Editor) and three reviewers, one of whom, Michael Paulin (Reviewer #3), served as a guest Reviewing Editor. The reviewers have discussed the reviews with one another and the Reviewing Editor has drafted this decision to help you prepare a revised submission.

Summary:

The manuscript brings new evidence to the hypothesis that cerebellum is directly involved in cognition, beyond as a consequence of a role in motor control and perceptual acuity. It argues that there is a correlation between expansion of lateral cerebellar lobules and cognitive abilities in mammalian evolution. The hypothesis is not new, but this manuscript introduces powerful modern methods of Bayesian inference to the debate. The study appears well executed and the manuscript is well written.

Essential revisions:

1) Are there differences in Purkinje cell numbers and/or folding of the cerebellum that also reflect species differences in behaviour? The scaling 'rules' presented by Herculano-Houzel and colleagues typically avoid discussing Purkinje cells because they are not quantified, but this is potentially an important metric for understanding the relationship between cognition and cerebellar anatomy. We do not expect the authors to undergo quantification of cerebellar folding and cell counts, but some discussion of the potential roles of cerebellar cortical folding and Purkinje cell numbers in relation to their data and conclusions is important to understanding the evolution of the cerebellum and the brain in general. It would also help to emphasize the points raised at the end of the Discussion concerning the importance of volumetric measurements in combination with cell counts rather than focusing exclusively on cell counts that ignore an entire population of cerebellar neurons.

2) The focus of the paper is on the relationship between the expansion of the lateral hemispheres and cognition, but there is very little discussion of cognition in pinnipeds or cetaceans. If the basis of this convergent/parallel evolution of the cerebellum is cognition, what evidence is there for superior cognitive abilities in cetaceans and pinnipeds? Realize that this can be a contentious issue and a lengthy discussion or review is not needed, but the authors need to provide some kind of evidence if the overarching theory is that the lateral expansion is related to cognition.

3) Obtaining data for datasets like these is difficult and time consuming, but the authors do not sufficiently discuss the potential effects of including other species in their analyses. For example, bats purportedly have an enlarged paraflocculus that is thought to be associated with echolocation. As such, they too could have enlarged lateral hemispheres relative to other taxa. Similarly, the inclusion of baleen whales could also potentially shift the data for odontocetes. Although the authors need not speculate on the cerebellar anatomy of these taxa or how that would specifically affect the analyses, some discussion of the taxonomic breadth of the current study is warranted as some of the results would likely change with the addition of other species. Currently, there is a paragraph in the Conclusions that discusses the need for greater taxonomic breadth, but it does not cover the specific examples I highlight here or reference to Larsell who did comment on species variation in cerebellar morphology across mammals. A paragraph that discusses bats, baleen whales and even the potential for other taxa (e.g., rodents, tree shrews, marsupials to name a few) to affect their calculations of evolutionary rates of change, key nodes in mammalian phylogeny and perhaps even the suggested link between cognition and lateral cerebellar expansion is needed to fully explain the limitations of the current study.

4) The trait evolution literature is littered with awfully thought out tests, often with incorrect mathematics. As well, many seem to forget that our models are necessarily unrealistic, and that any test result will be dominated by systematic error. The authors have done a commendable job choosing the analyses to perform, and deciding what, and what not, to infer from them. This was particularly the case when the main text is read in conjunction with the (excellent) discussion in the supplementary material. One result that may be problematic, however, is the reversible jump MCMC on regime shifts under the multiple OU. While not familiar with the software produced by (Uyeda and Harmon, 2014), the posterior probabilities of models are notoriously difficult to infer using reversible jump MCMC (If they weren't then researchers would not put so much effort into methods for estimating Bayes factors). However one reviewer spent some time working carefully through Uyeda and Harmon (2014) and they argue (using simulations) that the approach really does work for their particular model. Not sure if anything needs to be done on this point, but wanted to bring the discussion on this point to your attention.

5) The claim that "cerebellum underlies the domain-general automation of associative learning abilities" is a hypothesis (speculation) currently backed by limited circumstantial evidence. The authors fit their results to this conclusion uncritically. For example, the "cognitive" tasks in primates that are "… independent of perceptual bias and contextual confounds" largely require motor agility. The different genera represented in the sample have diverse lifestyles and sensory apparatus. As even the authors admit, if the lateral expansion of the cerebellum is unpacked at the level of lobules there is evidently something different going on in different lineages. It's true that there are major connections between cerebellar hemispheres and visual areas of cortex in primates, but no mention is made of the equal or stronger connections with somatosensory inputs and cortex. This evidence is consistent with the classical view of cerebellar involvement in motor aspects of visually guided movement. Mustelids are active predators, while zebu are domesticated herbivores that have been selected for docility. Species with whiskers and/or prehensile appendages have major somatosensory expansions of lateral hemispheres, while artiodactyls essentially have rocks at the end of their limbs and very limited somatosensory inputs to the lateral hemispheres. The authors have cherry-picked what they tell the reader about these animals. A correlation with general cognitive ability shows up only at the grossest level across phyla, and given the opportunities to subdivide cerebellar anatomy in multiple ways, the reported p-values for this effect are unimpressive (Actually it seems a little odd to do a Bayesian analysis and then report p-values). We don't want to argue that the classical 20th century view of cerebellum is correct, but we also urge caution in interpreting the data in such a restricted way. The authors should rewrite the Introduction and Discussion to place their results in a broader context. Such a paper could make an important contribution to the debate about cerebellar involvement in cognition.

---

## [Author Response]

Essential revisions:

1) Are there differences in Purkinje cell numbers and/or folding of the cerebellum that also reflect species differences in behaviour? The scaling 'rules' presented by Herculano-Houzel and colleagues typically avoid discussing Purkinje cells because they are not quantified, but this is potentially an important metric for understanding the relationship between cognition and cerebellar anatomy. We do not expect the authors to undergo quantification of cerebellar folding and cell counts, but some discussion of the potential roles of cerebellar cortical folding and Purkinje cell numbers in relation to their data and conclusions is important to understanding the evolution of the cerebellum and the brain in general. It would also help to emphasize the points raised at the end of the Discussion concerning the importance of volumetric measurements in combination with cell counts rather than focusing exclusively on cell counts that ignore an entire population of cerebellar neurons.

We agree that these are important discussion points that should be raised. We have adjusted the discussion accordingly (last paragraph of the Discussion section).

2) The focus of the paper is on the relationship between the expansion of the lateral hemispheres and cognition, but there is very little discussion of cognition in pinnipeds or cetaceans. If the basis of this convergent/parallel evolution of the cerebellum is cognition, what evidence is there for superior cognitive abilities in cetaceans and pinnipeds? Realize that this can be a contentious issue and a lengthy discussion or review is not needed, but the authors need to provide some kind of evidence if the overarching theory is that the lateral expansion is related to cognition.

We have now included a paragraph highlighting evidence for vocal production learning (VPL) in pinnipeds and cetaceans. VPL involves cerebellar-like error-based learning and has only been observed in a few species of mammals: humans, pinnipeds, dolphins, toothed whales, elephants and bats. Unfortunately, cerebellar hemispheric data on bats is currently not available (we have included a discussion on how such data may affect results; See comment 3.)

Except for the absence of bats in our sample, the occurrence of the capacity for VPL matches those species that are indicate by our analysis to have expanded their lateral cerebellum.

3) Obtaining data for datasets like these is difficult and time consuming, but the authors do not sufficiently discuss the potential effects of including other species in their analyses. For example, bats purportedly have an enlarged paraflocculus that is thought to be associated with echolocation. As such, they too could have enlarged lateral hemispheres relative to other taxa. Similarly, the inclusion of baleen whales could also potentially shift the data for odontocetes. Although the authors need not speculate on the cerebellar anatomy of these taxa or how that would specifically affect the analyses, some discussion of the taxonomic breadth of the current study is warranted as some of the results would likely change with the addition of other species. Currently, there is a paragraph in the Conclusions that discusses the need for greater taxonomic breadth, but it does not cover the specific examples I highlight here or reference to Larsell who did comment on species variation in cerebellar morphology across mammals. A paragraph that discusses bats, baleen whales and even the potential for other taxa (e.g., rodents, tree shrews, marsupials to name a few) to affect their calculations of evolutionary rates of change, key nodes in mammalian phylogeny and perhaps even the suggested link between cognition and lateral cerebellar expansion is needed to fully explain the limitations of the current study.

We are grateful to the reviewer for pointing out these examples. We now highlight these in the concluding paragraph that discusses the breadth of our sample.

4) The trait evolution literature is littered with awfully thought out tests, often with incorrect mathematics. As well, many seem to forget that our models are necessarily unrealistic, and that any test result will be dominated by systematic error. The authors have done a commendable job choosing the analyses to perform, and deciding what, and what not, to infer from them. This was particularly the case when the main text is read in conjunction with the (excellent) discussion in the supplementary material. One result that may be problematic, however, is the reversible jump MCMC on regime shifts under the multiple OU. While not familiar with the software produced by (Uyeda and Harmon, 2014), the posterior probabilities of models are notoriously difficult to infer using reversible jump MCMC (If they weren't then researchers would not put so much effort into methods for estimating Bayes factors). However one reviewer spent some time working carefully through Uyeda and Harmon (2014) and they argue (using simulations) that the approach really does work for their particular model. Not sure if anything needs to be done on this point, but wanted to bring the discussion on this point to your attention.

We couldn’t agree more and we appreciate that this point was raised. To overcome the uncertainties involved with the inference of posterior probabilities under reversible jump MCMC, we took the following steps:

1) We translated the r-j MCMC estimated model to a standard least-squares framework (i.c. pANCOVA) to test the significance of the estimated model. This overcomes the uncertainty of inferred posterior probabilities under a r-j MCMC model because it allows testing whether the estimated model given a certain posterior probability cutoff is significant. This procedure ensures an appropriate distinction between model estimation and model testing. The reversible jump MCMC was hereby used only for the purposes of estimating a best-fit model. This estimated model was then tested using pANCOVA.

2) It is, however, clear that pANCOVA does not provide an estimate of evolutionary history; it only allows testing the significance of the estimated variance patterning in the observed sample. To overcome the uncertainty of r-j OU MCMC estimation in modelling evolutionary history, we also computed rates of evolution using a multi-rate BM estimation (‘mvBM’). This multi-rate approach is a method that relies on different model assumptions than r-j OU MCMC modelling and can thus be regarded as an independent estimate of the same pattern. mvBM results confirm the exceptional rate of evolution in those ancestral lineages in which a regime-shift is inferred by r-j OU MCMC. The confirmation of the same pattern using different methods with different model assumption is highlights strong effect size and provides further support for our results.

We recognize that we did not sufficiently highlight these measures that we took to overcome the uncertainties involved in r-j OU MCMC modelling.

To highlight the above described measure 1), we have added a paragraph in the Materials and methods section entitled “Testing estimated changes in mean value”. This section follows sections on evolutionary modelling which are now entitle starting with “Estimating”. We hope that this sufficiently highlights the crucial difference between estimation, and testing the estimated patterns.

To highlight the above described measure 2), we have added a paragraph in the Materials and methods section entitled “Model uncertainty, reliability, and effect size”.

5) The claim that "cerebellum underlies the domain-general automation of associative learning abilities" is a hypothesis (speculation) currently backed by limited circumstantial evidence. The authors fit their results to this conclusion uncritically.

We fully recognize the cerebellum’s role in cognition is a working hypothesis that has limited evidence. We have rephrased the Introduction and Discussion as such.

For example, the "cognitive" tasks in primates that are "… independent of perceptual bias and contextual confounds" largely require motor agility. The different genera represented in the sample have diverse lifestyles and sensory apparatus. As even the authors admit, if the lateral expansion of the cerebellum is unpacked at the level of lobules there is evidently something different going on in different lineages. It's true that there are major connections between cerebellar hemispheres and visual areas of cortex in primates, but no mention is made of the equal or stronger connections with somatosensory inputs and cortex. This evidence is consistent with the classical view of cerebellar involvement in motor aspects of visually guided movement. Mustelids are active predators, while zebu are domesticated herbivores that have been selected for docility. Species with whiskers and/or prehensile appendages have major somatosensory expansions of lateral hemispheres, while artiodactyls essentially have rocks at the end of their limbs and very limited somatosensory inputs to the lateral hemispheres.

We fully agree. Though we do not see the cerebellum’s role in cognition as distinct from the role it plays in organizing models of motor behavior. Rather, we see the former as an extension of the latter. This argument was most recently made by Sokolov et al. (2017). Specifically, the argument is that, in animals that display high behavioral complexity, the manner in which motor models are internalized to produce smooth and automated movement was expanded to domains beyond motor function. This argument was perhaps not made explicit. We have added to our Introduction and Discussion to remedy this.

The authors have cherry-picked what they tell the reader about these animals.

We recognize the importance of somatosensory connections to the lateral cerebellum in pinnipeds and primates and thank the reviewer for drawing our attention to this as it further strengthens our conclusions. The reason it strengthens our conclusions is because the importance of vision in primates is intrinsically linked to the evolution of visual grasping and reaching. This importance is highlighted in the evolution of the dorsal visual stream (providing a specialized visuo-somatosensory pathway that integrates parietal, somatosensory, and motor processing). We have added these elements to our Discussion.

We further discuss the visual and somatosensory specializations of pinnipeds in more detail.

A correlation with general cognitive ability shows up only at the grossest level across phyla, and given the opportunities to subdivide cerebellar anatomy in multiple ways, the reported p-values for this effect are unimpressive (Actually it seems a little odd to do a Bayesian analysis and then report p-values).

We would like to clarify that the Bayesian analysis and the *P*-values that stem from least-squares analysis refer to separate parts of the research design. Please see response to comment 4.

The Bayesian analysis was not used for the purposes of correlating lateral cerebellar volumes to measures of cognition in primates. This was done using phylogenetic generalized least-squares analysis (resulting in a *P*-value).

We don't want to argue that the classical 20th century view of cerebellum is correct, but we also urge caution in interpreting the data in such a restricted way. The authors should rewrite the Introduction and Discussion to place their results in a broader context. Such a paper could make an important contribution to the debate about cerebellar involvement in cognition.

We would like to thank the reviewer for urging that we provide a more nuanced discussion of our results. We have adjusted our Introduction and Discussion accordingly.